# Your Transformer May Not be as Powerful as You Expect

**Shengjie Luo**[1,5]*, **Shanda Li**[2]*, **Shuxin Zheng**[3], **Tie-Yan Liu**[3], **Liwei Wang**[1,4]†, **Di He**[1]†

[1]National Key Laboratory of General Artificial Intelligence,
School of Intelligence Science and Technology, Peking University
[2]Machine Learning Department, School of Computer Science, Carnegie Mellon University
[3]Microsoft Research   [4]Center for Data Science, Peking University   [5]Zhejiang Lab
luosj@stu.pku.edu.cn, shandal@cs.cmu.edu,
{shuz, tyliu}@microsoft.com, {wanglw,dihe}@pku.edu.cn

## Abstract

Relative Positional Encoding (RPE), which encodes the relative distance between any pair of tokens, is one of the most successful modifications to the original Transformer. As far as we know, theoretical understanding of the RPE-based Transformers is largely unexplored. In this work, we mathematically analyze the power of RPE-based Transformers regarding whether the model is capable of approximating any continuous sequence-to-sequence functions. One may naturally assume the answer is in the affirmative—RPE-based Transformers are universal function approximators. However, we present a negative result by showing there exist continuous sequence-to-sequence functions that RPE-based Transformers cannot approximate no matter how deep and wide the neural network is. One key reason lies in that most RPEs are placed in the softmax attention that always generates a right stochastic matrix. This restricts the network from capturing positional information in the RPEs and limits its capacity. To overcome the problem and make the model more powerful, we first present sufficient conditions for RPE-based Transformers to achieve universal function approximation. With the theoretical guidance, we develop a novel attention module, called Universal RPE-based (URPE) Attention, which satisfies the conditions. Therefore, the corresponding URPE-based Transformers become universal function approximators. Extensive experiments covering typical architectures and tasks demonstrate that our model is parameter-efficient and can achieve superior performance to strong baselines in a wide range of applications. The code will be made publicly available at https://github.com/lsj2408/URPE.

## 1   Introduction

Transformer [61] is well acknowledged as a powerful neural network in modeling sequential data [12, 42]. Relative Positional Encoding (RPE) is one of the most successful modifications to the Transformer model [49]. Unlike the originally designed Absolute Positional Encoding (APE) that encodes each position as an embedding vector, RPE encodes the relative distance between any pair of tokens and is usually placed in the softmax exponentiation in the self-attention module. Empirically, many studies show that RPE-based Transformers can achieve impressive performance on various language tasks [54, 10] and have better extrapolation ability on longer sequences [52]. Another point worth noting is that RPE makes Transformer easily be extended to other data modalities,

---

*Equal contribution. The order is decided by rolling the dice.
†Correspondence to: Di He<dihe@pku.edu.cn>, Liwei Wang <wanglw@pku.edu.cn>.

36th Conference on Neural Information Processing Systems (NeurIPS 2022).

such as image [14, 43] and graph [67], as the relative distance naturally preserves invariant properties for several important transformations like rotation and translation.

In this paper, we first investigate the theoretical aspect of the RPE-based Transformers. In particular, we study their expressive power which describes models' ability to approximate any continuous functions. Recently, Yun et al. [69] proved that the APE-based Transformers are universal approximators of continuous sequence-to-sequence functions on a compact domain, and one may expect that the RPE-based Transformers enjoy the same property. However, we provide a surprising theoretical finding which shows that widely-used Transformers with RPE are *not* universal function approximators, i.e., there exist continuous sequence-to-sequence functions that the models cannot approximate no matter how deep and wide the model is. One key observation is that the RPEs are placed inside the softmax in the attention module. The softmax operator always generates a right stochastic matrix, which fails to reflect enough positional information encoded in RPE to the output. Synthetic tasks are conducted to support this mathematical claim.

To design a more powerful RPE-based Transformer, we delve into the limitation of the model and theoretically derive two sufficient conditions to achieve universal function approximation: the *attentive condition* and *position-aware condition*. Both conditions together state that the RPE-based attention function class should cover some special cases of the originally designed attention and break the right-stochastic-matrix limitation. With such theoretical guidance, we develop a new attention module called Universal RPE-based (URPE) Attention that satisfies the above conditions. Therefore, the Transformers with URPE-based Attention, called *URPE-based Transformers*, are universal function approximators. We show our proposed architecture is easy to implement and parameter-efficient via extensive experiments covering typical model architectures [54, 10, 67] and tasks (synthetic tasks, language modeling, and graph learning). Our model brings consistent performance gains compared with existing RPE-based Transformers on a wide range of tasks.

The paper is organized as follows. In Section 2, we introduce background on the Transformer architecture and positional encoding approaches. In Section 3, we prove that the widely-used RPE-based Transformers are *not* universal function approximators. In Section 4, we further present sufficient conditions for RPE-based Transformers to achieve universal approximation, and develop a new attention module, URPE-based Attention, to build a universal RPE-based Transformer. Experiments are presented in Section 5 to demonstrate the effectiveness of Transformers with our proposed URPE-based Attention. Related works and the conclusion are discussed in the last two sections.

## 2   Preliminary

The Transformer architecture is composed of stacked Transformer blocks [61, 12]. A Transformer block is a sequence-to-sequence mapping from $\mathbb{R}^{n \times d}$ to $\mathbb{R}^{n \times d}$, where $n$ is the sequence length and $d$ is the dimension of token embedding. A Transformer block consists of two layers: a self-attention layer followed by a feed-forward layer, with both layers having normalization (e.g., LayerNorm [1], RMSNorm [71]) and skip connections. For an input $\boldsymbol{X} \in \mathbb{R}^{n \times d}$, the self-attention layer and feed-forward layer are defined as follows:

$$\boldsymbol{A}^h(\boldsymbol{X}) = \mathrm{softmax}\left(\boldsymbol{X}\boldsymbol{W}_Q^h(\boldsymbol{X}\boldsymbol{W}_K^h)^\top\right); \tag{1}$$

$$\mathrm{Attn}(\boldsymbol{X}) = \boldsymbol{X} + \sum_{h=1}^{H} \boldsymbol{A}^h(\boldsymbol{X})\boldsymbol{X}\boldsymbol{W}_V^h\boldsymbol{W}_O^h; \tag{2}$$

$$\mathrm{FFN}(\boldsymbol{X}) = \boldsymbol{X} + \mathrm{ReLU}(\boldsymbol{X}\boldsymbol{W}_1)\boldsymbol{W}_2, \tag{3}$$

where $\boldsymbol{W}_O^h \in \mathbb{R}^{d_H \times d}$, $\boldsymbol{W}_Q^h, \boldsymbol{W}_K^h, \boldsymbol{W}_V^h \in \mathbb{R}^{d \times d_H}$, $\boldsymbol{W}_1 \in \mathbb{R}^{d \times r}$, $\boldsymbol{W}_2 \in \mathbb{R}^{r \times d}$. $H$ is the number of attention heads, $d_H$ is the dimension of each head, and $r$ is the dimension of the hidden layer. $\boldsymbol{A}^h(\boldsymbol{X})$ is usually referred to as the *attention matrix*. Given pre-defined $H$, $d_H$ and $r$, we refer to the function class of the Transformer blocks as T_blocks$(H, d_H, r)$.

**Transformer with Absolute Positional Encoding.** Self-attention layers and feed-forward layers defined in Eq.(2) and (3) are entirely invariant to sequence order. Therefore, purely stacked Transformer blocks cannot distinguish information at different positions. The original Transformer [61] proposes Absolute Positional Encoding (APE) to endow Transformer networks with the ability to

capture positional information. In particular, a (learnable) real-valued embedding $e_i \in \mathbb{R}^d$ is assigned to each position $i$, leading to an Absolute Positional Encoding matrix $\boldsymbol{E} = [e_1, \cdots, e_n]^\top$, which will be added to the input sequence. Formally speaking, the function class represented by APE-based Transformers is

$$\Omega_{\mathrm{APE}}^{H,d_H,r} = \{f(\boldsymbol{X}) = g(\boldsymbol{X} + \boldsymbol{E}) | \boldsymbol{E} \in \mathbb{R}^{n \times d}; g = g_L \circ \cdots \circ g_1; g_i \in \mathrm{T\_blocks}(H, d_H, r); L \in \mathbb{N}^*\}.$$

APE essentially enhances the expressive power of Transformers. Yun et al. [69] proved the following theoretical result, which shows that APE-based Transformers can approximate any continuous sequence-to-sequence function in a compact domain.

**Theorem 1** (informal [69]). *Given $n, d \in \mathbb{N}^*$, the function class of Transformers with APE, $\Omega_{\mathrm{APE}}^{2,1,4}$, is a universal approximator for continuous functions that map a compact domain in $\mathbb{R}^{n \times d}$ to $\mathbb{R}^{n \times d}$.*

Though Transformers with APE are conceptionally simple and enjoy good theoretical properties, they have a few known shortcomings. For example, Press et al. [52] showed that APE-based Transformers usually generalize poorly to longer sequences, as those positional embeddings for large indexes are hardly trained. Many works [58, 10, 54, 33, 26] employ Relative Positional Encoding (RPE), which becomes increasingly popular as a powerful way to encode positional information for Transformers and largely overcomes the disadvantages of APE.

**Transformer with Relative Positional Encoding.** Different from APE that assigns an embedding $e_i$ for each position $i$, Relative Positional Encoding (RPE) encodes relative distance $i - j$ for each position pair $(i, j)$. With the relative positional encoding, most previous works modified the attention computation defined in Eq.(1) as follows:

$$\boldsymbol{A}_{\mathrm{RPE}}^h(\boldsymbol{X}) = \mathrm{softmax}\left(\boldsymbol{X}\boldsymbol{W}_Q^h(\boldsymbol{X}\boldsymbol{W}_K^h)^\top + \boldsymbol{B}\right), \tag{4}$$

where $\boldsymbol{B}$ is an $n \times n$ matrix. The $(i, j)$-th entry of $\boldsymbol{B}$, denoted by $b_{ij}$, models the interaction between the $i$-th and $j$-th position. Different parameterizations of $\boldsymbol{B}$ lead to different model architectures. A few well-known examples include:

- Shaw's RPE [58]: $b_{ij} = \boldsymbol{X}_i \boldsymbol{W}_Q^h \boldsymbol{r}_{i-j}^\top$, where $\boldsymbol{r}_{i-j}$ are learnable vectors.
- T5 [54]: $b_{ij} = m_{i-j}$, where $m_{i-j}$ are learnable scalars, i.e., $\boldsymbol{B}$ is parameterized as a Toeplitz matrix [22, 45].
- DeBERTa [25]: $b_{ij} = \boldsymbol{X}_i \boldsymbol{W}_Q^h \boldsymbol{r}_{i-j}^\top + \boldsymbol{s}_{i-j}(\boldsymbol{X}_j \boldsymbol{W}_K^h)^\top$, where $\boldsymbol{r}_{i-j}$ and $\boldsymbol{s}_{i-j}$ are learnable vectors.
- Transformer-XL [10]: $b_{ij} = \boldsymbol{X}_i \boldsymbol{W}_Q^h (\boldsymbol{r}_{i-j} \tilde{\boldsymbol{W}}_K^h)^\top + \boldsymbol{u}(\boldsymbol{X}_j \boldsymbol{W}_K^h)^\top + \boldsymbol{v}(\boldsymbol{r}_{i-j} \tilde{\boldsymbol{W}}_K^h)^\top$, where $\boldsymbol{u}, \boldsymbol{v}$ and $\tilde{\boldsymbol{W}}_K^h$ are all learnable vectors/matrix, and $\boldsymbol{r}_{i-j}$ are sinusoidal positional encoding vectors fixed during training.

Several interesting phenomena suggest that RPE-based Transformers have many advantages compared to their APE-based counterparts. Press et al. [52] demonstrated that RPE-based Transformers generalize better on longer sequences. T5 [54] and Transformer-XL [10] show that Transformers with RPE can achieve strong performance in language understanding and language generation tasks. Recently, RPEs are also popularly used in other domains to encode translation/rotation-invariant structural signals. Typical examples include Swin Transformer [43] and Graphormer [67], both of which use RPE and achieve state-of-the-art performance in image and graph representation learning.

## 3 Transformers with RPE are not Universal Approximators

We are interested in the expressive power of Transformers with RPE and investigate whether this architecture is as powerful as the original APE-based Transformers. To make comparison, we similarly define the function class of the Transformer blocks with RPE-based attention (Eq.(4)) as $\mathrm{T\_blocks}_{\mathrm{RPE}}(H, d_H, r)$, in which the relative positional encoding matrix $\boldsymbol{B}$ is assumed to be an *arbitrary parameterized mapping* from the input $\boldsymbol{X}$ to an $n \times n$ matrix. The function class represented by Transformers with RPE is defined as:

$$\Omega_{\mathrm{RPE}}^{H,d_H,r} = \{g_L \circ \cdots \circ g_1 : \mathbb{R}^{n \times d} \to \mathbb{R}^{n \times d} | g_1, \cdots, g_L \in \mathrm{T\_blocks}_{\mathrm{RPE}}(H, d_H, r), L \in \mathbb{N}^*\}.$$

Surprisingly, we present a negative theoretical result: we prove that the function class of Transformers with RPE, $\Omega_{\mathrm{RPE}}^{H,d_H,r}$, is *not* a universal approximator for sequence-to-sequence functions.

**Theorem 2.** *Given $n > 2$, $d$ and $\mathcal{D} \subseteq \mathbb{R}^{n \times d}$, assume that the all-zero matrix $\mathbf{0} \in \mathcal{D}$. For any $M > 0$, there exists a continuous function $\tilde{g}_M : \mathcal{D} \to \mathbb{R}^{n \times d}$, such that*

$$\sup_{\boldsymbol{X} \in \mathcal{D}} \|\tilde{g}_M(\boldsymbol{X}) - g(\boldsymbol{X})\|_F > M \tag{5}$$

*holds for any $g \in \Omega_{\mathrm{RPE}}^{H,d_H,r}$, where $H, d_H, r \in \mathbb{N}^*$.*

*Proof.* Without loss of generality, we prove the theorem for $d = 1$. The proof can be easily extended to $d > 1$ settings. Given $M > 0$, we consider a specific sequence-to-sequence function as target: $\tilde{g}_M : \boldsymbol{X} \mapsto (2M, 0, \cdots, 0)^\top$. To show $\sup_{\boldsymbol{X} \in \mathcal{D}} \|\tilde{g}_M(\boldsymbol{X}) - g(\boldsymbol{X})\|_F > M$ holds for any $g \in \Omega_{\mathrm{RPE}}^{H,d_H,r}$, we pick up an input $\boldsymbol{X}^*$, which is composed of $n$ *identical* row vectors in $\mathbb{R}^d$, i.e., the sequence consists of $n$ identical tokens. Since the function $\boldsymbol{A}_{\mathrm{RPE}}^h(\boldsymbol{X})$ outputs a right stochastic matrix, it is easy to check that $\mathrm{Attn}(\boldsymbol{X}) = \boldsymbol{X} + \sum_{h=1}^H \boldsymbol{A}^h(\boldsymbol{X}) \boldsymbol{X} \boldsymbol{W}_V^h \boldsymbol{W}_O^h$ is also composed of $n$ *identical* row vectors in $\mathbb{R}^d$. Note that $\mathrm{FFN}(\boldsymbol{X})$ and normalizations operate identically on each row vector, we can obtain that the final output of a Transformer block with RPE-based attention is still composed of $n$ *identical* row vectors in $\mathbb{R}^d$.

Since $g$ is a composition of multiple Transformer blocks in $\mathrm{T\_blocks}_{\mathrm{RPE}}(H, d_H, r)$ and $\mathbf{0}$ is composed of $n$ *identical* row vectors in $\mathbb{R}^d$, we conclude from the analysis above that $g(\mathbf{0})$ is also composed of $n$ *identical* row vectors in $\mathbb{R}^d$, i.e., there exists $c \in \mathbb{R}^d$ such that $g(\mathbf{0}) = c\mathbf{1}_n^\top$. Therefore, by applying Cauchy-Schwartz Inequality we obtain

$$\|\tilde{g}_M(\mathbf{0}) - g(\mathbf{0})\|_F^2 = (2M - c)^2 + (n-1)c^2 \geq \frac{4M^2}{1 + \frac{1}{n-1}} > M^2 \Rightarrow \sup_{\boldsymbol{X} \in \mathcal{D}} \|\tilde{g}_M(\boldsymbol{X}) - g(\boldsymbol{X})\|_F > M,$$

which completes the proof. $\qquad\square$

**Discussions.** The key observation in Theorem 2 is that $\boldsymbol{A}_{\mathrm{RPE}}(\boldsymbol{X})$ always outputs a right stochastic matrix. Even if RPE carries rich positional information, such signal will be suppressed to satisfy $\boldsymbol{A}_{\mathrm{RPE}}(\boldsymbol{X})\mathbf{1} = \mathbf{1}$ for arbitrary $\boldsymbol{X} \in \mathbb{R}^{n \times d}$, where $\mathbf{1}$ is an all-one n-dimensional vector. As a result, the attention module fails to reflect enough positional information encoded in RPE to the output, which restricts the model capacity. The problem will be significant when the target function $\tilde{g}$ is very position-sensitive (in the extreme case, $\tilde{g}$ only depends on the position indexes). We also conduct experiments on simple sequence-to-sequence tasks using synthetic data to support this mathematical claim in Section 5.1. One may expect that simply removing the denominator in the $\mathrm{softmax}$ can break the limitation. However, this modification brings significant optimization instability in practice. Given that RPE has many advantages[3] compared to APE, it's appealing to design an RPE-based Transformer variant that is a universal approximator of sequence-to-sequence functions and easy to optimize. This is what we precisely work on in the next section.

## 4 Making RPE-based Transformers Universal Approximators

This section contains two sub-sections. In the first sub-section, we provide a sufficient condition for the RPE-based Transformers to achieve universal approximation. In the second sub-section, we offer a practical instantiation of the Transformer with RPE that satisfies the requirement and is parameter-efficient.

---

[3]On one hand, we claim that RPE-based Transformers cannot achieve universal function approximation (while APE-based models can achieve). On the other hand, we claim RPE is empirically advantageous compared to APE, according to previous works. One may feel there exists a contradiction and get confused. We would like to clarify that the two claims are made in different settings. For example, RPE empirically performs well in generalization to longer sequences, while the theoretical analysis of approximation capability focuses on functions with bounded input lengths (See Theorem 1 where $n$ is given). Our goal is to design an RPE variant with the same expressiveness as APE in the theoretical setting and enjoys its usefulness in practical scenarios.

### 4.1 A Sufficient Condition to Achieve Universal Approximation

Motivated by formulation (1) and (4), we consider a general form $\boldsymbol{A}_U^h : \mathbb{R}^{n \times d} \to \mathbb{R}^{n \times n}$ and define the corresponding attention layer as

$$\text{Attn}_U(\boldsymbol{X}) = \boldsymbol{X} + \sum_{h=1}^{H} \boldsymbol{A}_U^h(\boldsymbol{X})\boldsymbol{X}\boldsymbol{W}_V^h \boldsymbol{W}_O^h, \tag{6}$$

We further define the function class of the corresponding Transformer block as $\text{T\_blocks}_U(H, d_H, r)$, and define the function class of Transformers composed of stacked $\text{T\_blocks}_U(H, d_H, r)$ as:

$$\Omega_U^{H,d_H,r} = \{g_L \circ \cdots \circ g_1 : \mathbb{R}^{n \times d} \to \mathbb{R}^{n \times d} \mid g_1, \cdots, g_L \in \text{T\_blocks}_U(H, d_H, r), L \in \mathbb{N}^*\}. \tag{7}$$

Our goal is to investigate the requirements on $\boldsymbol{A}_U^h$ under which the induced function class $\Omega_U^{H,r}$ can become universal approximators of continuous sequence-to-sequence functions. We provide one sufficient condition in the following theorem. Following Yun et al. [69] and many other previous theoretical works [44, 24, 70], we study the expressiveness of a simplified version of Transformer in which normalization layers are omitted, as it is widely believed that normalization mainly helps optimization but does not hurt the expressive power of the network [32, 1].

**Theorem 3.** *Given $n, d \in \mathbb{N}^*$, $p \in [1, +\infty)$, $\varepsilon > 0$, a compact set $\mathcal{D} \subseteq \mathbb{R}^{n \times d}$, and a continuous sequence-to-sequence function $f : \mathcal{D} \to \mathbb{R}^{n \times d}$. Assume that $\boldsymbol{A}_U^h$ satisfies the following conditions:*

- ***Attentive condition.*** *For any $\boldsymbol{u} \in \mathbb{R}^{d \times 1}$ and $c \in \mathbb{R}$, there exists a parametrization of $\boldsymbol{A}_U^h$, such that $\boldsymbol{A}_U^h(\boldsymbol{X}) = \text{softmax}\left(\boldsymbol{X}\boldsymbol{u}(\boldsymbol{X}\boldsymbol{u} - c\boldsymbol{1})^\top\right)$.*

- ***Position-aware condition.*** *There exists a parametrization of $\boldsymbol{A}_U^h$ and a vector $\boldsymbol{v} \in \mathbb{R}^n$ whose entries are all distinct, such that $\boldsymbol{A}_U^h(\boldsymbol{X})\boldsymbol{1} = \boldsymbol{v}$ for any $\boldsymbol{X} \in \mathbb{R}^{n \times d}$.*

*Then there exists a Transformer network $g \in \Omega_U^{2,1,4}$ such that $\left(\int_{\mathcal{D}} \|f(\boldsymbol{X}) - g(\boldsymbol{X})\|_p^p \mathrm{d}\boldsymbol{X}\right)^{\frac{1}{p}} < \varepsilon$, where $\|\cdot\|_p$ denotes the entry-wise $\ell^p$ norm for matrices.*

The detailed proof of Theorem 3 can be found in Appendix A. Theorem 3 presents two conditions to make RPE-based Transformers become universal approximators. Intuitively, the *attentive condition* states that $\boldsymbol{A}_U^h$ should contain a special case of the original attention matrix in Eq.(1), where $\boldsymbol{W}_Q = \boldsymbol{W}_K \in \mathbb{R}^{d_H \times d}$ and $d_H = 1$. The *position-aware condition* states that $\boldsymbol{A}_U^h$ needs to break the limitation of $\boldsymbol{A}(\boldsymbol{X})$ being a right stochastic matrix (i.e., $\boldsymbol{A}(\boldsymbol{X})\boldsymbol{1} = \boldsymbol{1}$ for all $\boldsymbol{X} \in \mathbb{R}^{n \times d}$). We will present a concrete example satisfying these conditions in the next subsection.

### 4.2 A Universal RPE-based Transformer

We develop a new Transformer variant which satisfies the conditions above. In particular, we multiply the softmax attention matrix with another matrix, and obtain

$$\boldsymbol{A}_U(\boldsymbol{X}) = \text{softmax}\left(\boldsymbol{X}\boldsymbol{W}_Q(\boldsymbol{X}\boldsymbol{W}_K)^\top + \boldsymbol{B}\right) \odot \boldsymbol{C}, \tag{8}$$

where $\odot$ denotes entry-wise product. $\boldsymbol{B}$ can take any form described in Section 2. We refer to this attention variant as **Universal RPE-based Attention** (URPE-based attention).

To make URPE-based attention rely relative positional information, we set $\boldsymbol{C} \in \mathbb{R}^{n \times n}$ to be a learnable Toeplitz matrix in which each element on the descending diagonal from left to right has the same value. Note that a Toeplitz matrix of shape $n \times n$ has $2n - 1$ degrees of freedom. Therefore, we only need $2n - 1$ new parameters for each $\boldsymbol{C}$. It can be proved that URPE-based Attention satisfies the two conditions in Theorem 3 and the proof can be found in Appendix B.

**Proposition 4.** *URPE-based Attention defined in Eq.(8) satisfies the conditions in Theorem 3. Consequently, given $n, d \in \mathbb{N}^*$, Transformers using this form of attention are universal approximators of continuous sequence-to-sequence functions that map a compact domain in $\mathbb{R}^{n \times d}$ to $\mathbb{R}^{n \times d}$.*

To further improve parameter efficiency, we set $\boldsymbol{C}$ to be shared across different layers but to be unique for each attention head. As an example, in Transformer-XL (see Section 5.2), we introduce only about

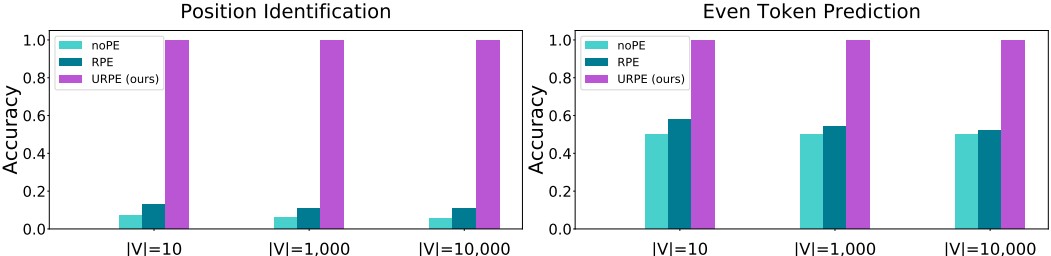

Figure 1: Results on synthetic sequence-to-sequence tasks: (1) Position Identification (Left Panel); (2) Even Token Prediction (Right Panel). $|V|$ is the vocabulary size. The URPE-based Transformer model consistently solves both tasks across different settings while other methods fail.

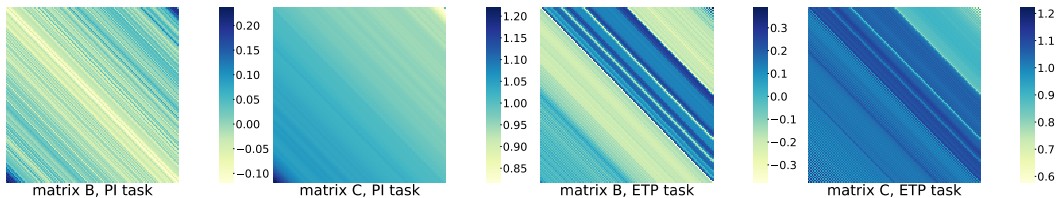

Figure 2: Visualizations of the learned Universal RPE (matrix $B$ and $C$ in Eq.(8)). It can be easily seen that the matrix $B$ and $C$ capture different aspects of positional information.

4K new parameters, which is negligible compared to the 151M parameters of the Transformer-XL model but still leads to non-trivial improvements.

We make two more discussions for our designed URPE-based attention. The first one is about applying URPE-based attention in the causal setting. Causal attention is important in language generation tasks. URPE-based Attention is compatible with it as one can set the $(i, j)$-th entry of $C$ to be 0 for $i > j$. The second one is on the initialization of $C$. In practice, the matrix $C$ can be initialized as an all-one matrix. Therefore, the model behaves identically to the original RPE-based model at initialization, and the additional parameters in $C$ are learned gradually. We can also fine-tune any well-trained RPE-based Transformer to its URPE-based counterpart by setting $C$ as an all-one matrix and further fine-tuning the model to learn $C$.

## 5 Experiments

In this section, we empirically study the effectiveness of the proposed model. In particular, we aim at answering the following questions through experiments:

- **Question 1**: Can the theoretical results on the approximation capability of RPE-based Transformer and URPE-based Transformer be reflected in certain experiments?
- **Question 2**: With different RPE methods (the matrix $B$ in Eq.(8)), can URPE-based Transformer outperform its RPE-based counterpart in real-world applications?
- **Question 3**: Can URPE-based Attention serve as a versatile module to improve the general Transformers beyond language tasks?

We will answer each question with carefully designed experiments in the following sub-sections. Due to space limitation, we only present results on representative model architectures, task types, and data modalities in the main body of the paper. More results are presented in the appendix. All codes are implemented based on the official codebases of Fairseq [50] and Graphormer [67] in PyTorch [51].

### 5.1 Synthetic Tasks

To empirically verify our theoretical results on the approximation capability of RPE-based Transformer and URPE-based Transformer, we design two synthetic tasks: 1) Position Identification; 2) Even Token Prediction.

Table 1: Language model perplexity scores on WikiText-103 validation and test set. We use $^*$ to indicate the best performance. All the results of the baseline methods are reported in [10]

| Model | #Params | Valid Perplexity | Test Perplexity |
|---|---|---|---|
| LSTM [21] | - | / | 48.7 |
| TCN [2] | - | / | 45.2 |
| GCNN-8 [11] | - | / | 44.9 |
| LSTM+Neural cache [21] | - | / | 40.8 |
| GCNN-14 [11] | - | / | 37.2 |
| QRNN [46] | 151M | / | 33.0 |
| Hebbian+Cache [53] | - | / | 29.9 |
| Transformer-XL Base [10] | 151M | 23.1 | 24.0 |
| Transformer-XL Base + URPE-based Attention (ours) | 151M | 22.4$^*$ | 23.2$^*$ |

Both Position Identification (PI) task and Even Token Prediction (ETP) task are sequence-to-sequence prediction tasks. Given a sequence of tokens $s = (w_1, w_2, \cdots, w_n)$, the PI task is to predict the position index of each token in the sequence, i.e., the target sequence-to-sequence function $f$ to approximate can be defined as

$$f_{\mathrm{PI}}(w_1, w_2, \cdots, w_n) = (1, 2, \cdots, n) \tag{9}$$

The ETP task is defined as follows: for the first half of positions in a sequence, the task requires the model to output the input tokens at positions with even number index; for the remaining half of positions, the task requires the model to output the special token End-Of-Sentence (EOS), i.e.,

$$f_{\mathrm{ETP}}(w_1, w_2, \cdots, w_n) = (w_2, w_4, \cdots, w_n, \mathrm{EOS}, \cdots, \mathrm{EOS}) \tag{10}$$

Both tasks require the model to accurately encode the positional information, which would be difficult for RPE-based Transformers to capture. For both tasks, we use synthetic datasets with randomly generated sequences. In detail, we vary the token vocabulary size from [10, 1000, 10000] and set the sequence length to 128. We choose the vanilla Transformer as the base model and compare the following ways to encode positional information: 1) no positional encodings (noPE); 2) T5-style relative positional encoding (RPE) [54]; 3) URPE with T5-style RPE backbone Transformer. The number of layers and the number of attention heads are set to 3 and 12, respectively. The hidden dimension is set to 768.

**Results.** We use token-level accuracy as the evaluation metric. The experimental results are shown in Figure 1. From the figure, it can be easily seen that the Transformer without PE and the Transformer with T5-style RPE cannot perfectly solve the synthetic tasks (less than 60% accuracy). On the contrary, the URPE-based Transformer achieves $100\%$ accuracy on both tasks. Firstly, this result clearly indicates that our proposed model outperforms the backbone T5-style RPE-based Transformer by a large margin. Furthermore, we can see that even for such simple tasks, the Transformer with T5-style RPE sometimes fails, while the URPE-based Transformer succeeds and approximates the target function well, which is consistent with our theoretical findings. Lastly, we provide visualizations of the learned Universal RPE (Eq.(8)) on both tasks in Figure 2, which show that the matrix $B$ and $C$ capture different aspects of positional information.

## 5.2 Language Modeling

We use language modeling to study the effectiveness of the proposed URPE-based Attention. Language modeling is an important practical application which usually requires the modelling of long-term dependency between tokens. We conduct experiments on the WikiText-103 dataset [47], which contains 103M training tokens from 28K articles, with an average length of 3.6K tokens per article. Relative positional encoding methods are popularly used in language modelling. We choose Transformer-XL model [10] as the backbone model of our URPE-based Transformer. Following [10], the number of layers and the number of attention heads are set to 16 and 10 respectively. The dimension of hidden layers and feed-forward layers are set to 410 and 2100. The detailed descriptions of the baselines and training settings are presented in the appendix.

Table 2: Mean Absolute Error (MAE) on ZINC test set. We use * to indicate the best performance.

| Model | #Params | Test MAE on ZINC-Subset | Test MAE on ZINC-Full |
|---|---|---|---|
| GIN [66] | 509,549 | 0.526±0.051 | 0.088±0.002 |
| GraphSAGE [23] | 505,341 | 0.398±0.002 | 0.126±0.003 |
| GAT [62] | 531,345 | 0.384±0.007 | 0.111±0.002 |
| GCN [35] | 505,079 | 0.367±0.011 | 0.113±0.002 |
| MoNet [48] | 504,013 | 0.292±0.006 | 0.090±0.002 |
| GatedGCN-PE [5] | 505,011 | 0.214±0.006 | - |
| MPNN(sum) [20] | 480,805 | 0.145±0.007 | - |
| HIMP [17] | 614,516 | 0.151±0.006 | 0.036±0.002 |
| PNA [8] | 387,155 | 0.142±0.010 | - |
| GT [15] | 588,929 | 0.226±0.014 | - |
| SAN [37] | 508,577 | 0.139±0.006 | - |
| Graphormer [67] | 489,321 | 0.122±0.006 | 0.052±0.005 |
| Graphormer+URPE-based Attention (ours) | 491,737 | 0.086±0.007* | 0.028±0.002* |

Table 3: Results on PCQM4M from OGB-LSC. We use * to indicate the best performance. The results of the baselines are reported in [67, 29].

| Model | #Params | Valid MAE |
|---|---|---|
| GCN [35] | 2.0M | 0.1691 |
| GIN [66] | 3.8M | 0.1537 |
| GCN-VN [35, 20] | 4.9M | 0.1485 |
| GIN-VN [66, 20] | 6.7M | 0.1395 |
| GINE-VN [6, 20] | 13.2M | 0.1430 |
| DeeperGCN-VN [38, 20] | 25.5M | 0.1398 |
| GT [15] | 0.6M | 0.1400 |
| GT-Wide [15] | 83.2M | 0.1408 |
| Graphormer [67] | 12.5M | 0.1264 |
| Graphormer + URPE-based Attention (ours) | 12.5M | 0.1238* |

**Results.** We show the perplexity scores on both validation and test set of different models in Table 1. It can be easily seen that the Transformer-XL equipped with our URPE-based attention achieves 22.4 and 23.2 valid and test perplexity scores, respectively, which are 0.7 and 0.8 lower than the backbone Transformer-XL model and also significantly better than other baselines. First, the results suggest that our proposed URPE-based attention can be well applied to Transformer-XL in real-world applications. Together with the observations in Section 5.1, we believe our URPE can be used in other RPE-based architectures, such as [58, 26]. It is worth noting that our model has negligible more parameters (about 4k) compared to the backbone Transformer-XL. Thus, the improvement in the perplexity should be mostly attributed to the stronger expressiveness of the model.

## 5.3 Graph Learning

We further examine whether the proposed URPE-based Attention can serve as a versatile module to improve the general RPE-based Transformers beyond language tasks. The Transformer-based models have become increasingly popular in the graph learning area [67, 37, 15]. Among those models, the recently proposed Graphormer [67] achieves state-of-the-art performance in many graph learning tasks [68, 59]. In Graph Transformers, RPE is used rather than APE since RPE only calculates distances between nodes, which naturally preserves many invariant and equivariant properties.

The attention computation in Graphormer also follows the Eq.(4) in Section 2. Specifically, Graphormer calculates the shortest-path distance between any pair of nodes and encodes this information as a bias term in the softmax attention to reflect the relative position of any node in the

Table 4: Comparison of RPE-based and UPRE-based Transformer-XL models of different sizes. We report perplexity scores on WikiText-103 validation set. $L$ denotes the number of layers.

| Model | $L = 4$ | $L = 8$ | $L = 16$ |
|---|---|---|---|
| Transformer-XL | 29.6 | 26.0 | 23.1 |
| Transformer-XL + URPE-based Attention (ours) | 28.7 | 25.2 | 22.4 |

Table 5: Inference Runtime (ms in log base 2) and Peak Memory Usage (GB) of RPE-based Transformer and URPE-based Transformer. $N$ denotes the input sequence length.

| Model | Inference Runtime | | | Peak Memory Usage | | |
|---|---|---|---|---|---|---|
| | $N = 128$ | $N = 256$ | $N = 512$ | $N = 128$ | $N = 256$ | $N = 512$ |
| RPE-based Transformer | 4.55 | 5.60 | 6.79 | 0.96 | 1.12 | 1.86 |
| URPE-based Transformer (ours) | 4.59 | 5.66 | 6.91 | 0.97 | 1.17 | 2.04 |

graph. We refer the readers to [67] for the detailed description of Graphormer. Similar to previous experiments, we adapt the URPE-based Attention to the Graphormer and compare them on two benchmark datasets covering graph representation learning tasks from small to large scale datasets: ZINC from Benchmarking-GNNs [16] and PCQM4M from Open Graph Benchmark Large Scale Challenge (OGB-LSC) [29]. For both tasks, we choose several competitive Transformer based models and GNNs as our baselines. Details of the experimental settings are presented in the appendix.

**ZINC.** ZINC is a real-world dataset which consists of 250K molecular graphs. The task is to predict the constrained solubility of a molecule which is an important chemical property for drug discovery. We train our models on both the ZINC-Full and ZINC-Subset (12K selected graphs following [16]). To demonstrate the power of our method and for fair comparison, we set the parameter budget of the model to be less than 500K following [16, 67]. We build on the Graphormer [67] model which consists of 12 layers. The dimension of hidden layers and feed-forward layers are set to 80. The number of attention heads are set to 32.

**PCQM4M.** PCQM4M is a quantum chemistry regression task in OGB-LSC [29]. The PCQM4M dataset contains more than 3.8 million molecular graphs in total, which is currently the largest graph-level prediction dataset. The state-of-the-art architecture for this task is the Graphormer model introduced above. We still follow [67] to set the hyper-parameters in the Graphromer model and equip it with URPE. In detail, our Graphormer with URPE-based Attention consists of 6 layers and 32 attention heads. The dimension of hidden layers and feed-forward layers are set to 512.

**Results.** The experimental results on ZINC and PCQM4M are shown in Table 2 and 3, where the score is averaged over four experiments with different seeds. It can be easily seen that the Graphormer model equipped with our URPE-based Attention consistently outperforms the backbone Graphormer model on both ZINC and PCQM4M tasks. In particular, our URPE-based Attention enables the Graphormer model to reduce more than 40% relative mean absolute error on the test set of ZINC-Subset and ZINC-Full. On the PCQM4M task, the improvement is around 0.003 mean absolute error which is also a significant improvement under the quantum chemistry precision. It is worth noting that our Graphormer model with URPE-based Attention achieves competitive performance compared to the Graphormer Base model with 48.3M parameters reported in [67]. Therefore, we believe our proposed architecture significantly improves the expressive power of the Transformer backbones and can be well extended to practical scenarios beyond language tasks.

### 5.4  More Analyses

**URPE-based Attention consistently improves the performance of models of different sizes.** We conduct ablation experiments on the Language Modeling task and vary the number of layers in [4, 8, 16] to investigate different model sizes. Following the experiment settings in Section 5.2, the number of attention heads is set to 10. The hidden dimension is set to 41. The dimension of the feed-forward layer is set to 2100. The results are presented in Table 4. It can be easily seen that our URPE-based Attention consistently reduces the perplexity scores of Transformer-XL models of different sizes, which indeed demonstrates the versatility of our URPE-based Attention.

**Runtime and Memory Usage Evaluation.** We further conduct memory and time costs profiling experiments on our URPE-based Transformers. We choose the vanilla Transformer as the backbone model. The number of layers and the hidden dimension are set to 12 and 768 respectively. The number of attention heads is set to 12. The batch size is set to 32. We vary the sequence length from [128, 256, 512]. We run profiling of all the models on a 16GB NVIDIA Tesla V100 GPU. Following Combiner [55], we compare the inference speed and memory costs of the vanilla Transformer with RPE and our URPE. The results are presented in Table 5, which show that our URPE only increases minor computational costs.

**Summary.** In this section, we design a series of experiments to answer the questions on the effectiveness and applicability of the proposed URPE-based Attention. All the experimental results suggest that our theoretical findings are convincing, and Transformers with our modification are effective, powerful, and widely applicable across different tasks.

## 6  Related Work

**Expressive power of Neural Networks.** Quantifying the capacity of neural networks is an important research direction in the literature on deep learning. [9, 19, 27, 4] showed that a neural network with one hidden layer and unbounded width can approximate arbitrary continuous functions on compact support with arbitrarily small error. Many works also studied the width efficiency on the expressive power of neural networks [44, 24, 40] and proved that ReLU networks with a bounded width but unlimited depth can achieve universal function approximation. Recently, there has been increasing interest in the theoretical understanding of Transformer models. Yun et al. [69] theoretically showed that Transformers can approximate any continuous sequence-to-sequence functions (i.e., universal approximation) on a compact domain by proving that stacking of self-attention layers can compute contextual mappings of the input embeddings. Dong et al. [13] analyzed the limitations of attention-only Transformers without considering FFN blocks, normalizations, and skip connections. Hron et al. [28] analyzed the behavior of multi-head attention and connect it with the Gaussian process when the number of heads tends to be infinity. In [7], it is proved that a multi-head attention layer with enough heads is at least as expressive as any convolution layer. All works above consider Transformers with absolute positional encoding, which is the main difference between them and our work.

**Positional encoding methods in Transformers.** In [61], the vanilla Transformer encodes the positional information via the absolute positional encoding (APE). Shaw et al. [58] is the first to introduce relative positional encoding (RPE) to Transformer. From then on, many works explored different RPE strategies based on [58]. Transformer-XL [10] re-parameterizes the self-attention to integrate relative positional encoding and enables long sequence modelling. T5 [54] simplifies the vector representations of relative positions to scalars. Kitaev et al. [36] disentangles the positional and content information in the Transformer encoder, which leads to an improved constituency parser. Ke et al. [33] further shows that such disentanglement also improves Transformer in general language pre-training and achieves superior performance on various downstream tasks. There are also works that encodes the positional information via other tools like trees [60], complex numbers [63], dynamic systems [41], Fourier features [39]. Compared to most previous works inspired by practical scenarios, our URPE-based attention is theoretically motivated by investigating the expressive power of RPE-based Transformers in a principled way.

## 7  Conclusion

In this paper, we first investigate the theoretical aspect of the RPE-based Transformers. In particular, we study their expressive power and provide a surprising theoretical finding which shows that widely-used Transformers with RPE are *not* universal function approximators. To design a more powerful RPE-based Transformer, we present sufficient conditions on the attention module to achieve universal function approximation and develop a novel *Universal RPE-based Transformer* using a new relative positional encoding approach. We conduct carefully designed experiments on synthetic sequence data, natural language, and graphs to show that our model brings consistent performance gains compared with existing RPE-based Transformers. In the future, we will benchmark our Universal RPE on other RPE-based Transformers and typical tasks to further verify its versatility as a basic module for Transformer-based models.

## Acknowledgements

We thank Tianle Cai and Jianlin Su for the helpful discussions. We also thank all the anonymous reviewers for the very careful and detailed reviews as well as the valuable suggestions. Their help has further enhanced our work. This work is supported by National Science Foundation of China (NSFC62276005), The Major Key Project of PCL (PCL2021A12), Exploratory Research Project of Zhejiang Lab (No. 2022RC0AN02), and Project 2020BD006 supported by PKUBaidu Fund.

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
