# OpenReview forum: "Your Transformer May Not be as Powerful as You Expect"
_NeurIPS.cc/2022/Conference — NeurIPS 2022 Accept_

### Official Review · Reviewer_hYM6 · 2022-07-07

**Rating:** 6
**Confidence:** 1
**Soundness:** 3 good
**Presentation:** 3 good
**Contribution:** 3 good

**Summary:**

The authors propose present an analysis showing that Transformers with Relative Position Encodings (RPE) are not universal function approximations and provide an alternative formulation, namely URPE, that satisfies the conditions for universal function approximation. The authors further perform a small empirical study showing that the proposed method outperforms baseline RPE both on synthetic tasks as well as real world datasets.

**Questions:**

Would it be possible for the authors to visualize the learned relative position embeddings? This tends often tends to be a good indicator to get an impression for what the position embeddings are actually learning.

**Limitations:**

As far as I can tell, the authors have not addressed potential limitations, but no limitations are apparent.

**Strengths And Weaknesses:**

Transformers build the foundation of many modern applications both in the research as well as the industrial community. As such improving the theoretical understanding and underpinning of those popular architecture is of great importance to the research community.

I appreciate that the analysis is presented in a clear and concise manner which makes it easy for the reader to follow.

I cannot come up with obvious weaknesses, however with the caveat that I am not very familiar with the underlying theory.

---

> ### Author Response · Authors · 2022-08-01
> **Response to Reviewer hYM6**
>
> Thank you very much for supporting our work! Due to space limitations, in our submission, we put the visualizations of the learned positional encoding in Appendix C.1 (See the supplementary material). It can be seen from the figures that the matrices $B$ and $C$ in the URPE capture different aspects of the positional information. We will consider moving these visualizations to the main body in the next version of the paper.
>
> We hope our responses above can address your questions and concerns, and we sincerely hope the reviewer can reevaluate our paper based on our responses.

---

### Official Review · Reviewer_BbEH · 2022-07-08

**Rating:** 7
**Confidence:** 3
**Soundness:** 4 excellent
**Presentation:** 4 excellent
**Contribution:** 4 excellent

**Summary:**

This paper shows the Transformer with Relational Positional Encoding (RPE) is not a universal function approximator theoretically and empirically. Thereafter, it presents sufficient conditions to be a universal function approximator and proposes a new method that achieve them, Universal RPE (URPE). It shows that URPE-based Transformers predict the position more accurately than Transformer with RPE and outperform the Transformer with other RPE on a wide range of applications.

============================

Great paper, and they addressed my concern enough. I'd like to sustain my score.

**Questions:**

- In Figure 1, it is weird RPE-based Transformers fail to predict the position when the number of tokens is 10. I expected that when the context length is short, RPE and URPE work fine and as the length increases, RPE fails. Could you explain this?


**Limitations:**

It proves it's argument (RPE-based Transformer is not a universal function approximator) and validates it's model URPE-based Transformer outperforms RPE-based Transformer for synthetic tasks and language modeling and graph learning. I think they addressed what they must consider so I don't have any concerns about the limitations over that.

**Strengths And Weaknesses:**

Strengths:
- Even though when $d=1$, why Transformer with RPE is not universal approximator is shown theoretically.
- It presents conditions to be a universal function approximator.
- It proposes URPE. URPE-based Transformer can cover the conditions suggested to be a universal function approximator.
- It evaluates URPE-based Transformers for a variety of tasks and transformer types (e.g., TrXL, Graphormer) and they show better performance than the Transformers without URPE.

Weaknesses:
- I couldn't find the critical weakness even though I tried to find by reading the manuscripts several times with references.

---

> ### Author Response · Authors · 2022-08-01
> **Response to Reviewer BbEH**
>
> Thanks very much for your appreciation of our work!
>
> The difference among groups in Figure 1 is the vocabulary size, not the sequence length. For all the synthetic data experiments, we set the sequence length to 128 and varied the token vocabulary size from [10, 1000, 10000]. See Line 231-233 for the experimental setting. Thus, Figure 1 indicates even when the vocabulary size grows (i.e., the task is more difficult), our URPE-based model can still approximate the target functions.
>
> It is apparent that RPE-based Transformers will get a higher accuracy for synthetic tasks with shorter sequence lengths. But different from our URPE-based model, it still cannot reach perfect performance according to Theorem 2. We are willing to add such experiments in the next version of the paper if you think those empirical results can improve the quality of the paper.

---

> > ### Comment · Reviewer_BbEH · 2022-08-05
> > **Thank you authors for your response**
> >
> > Hi authors,
> >
> > thank you for your response for my comments.
> >
> > I misunderstood the vocabulary size, yes, it is clear that when increasing the number of vocabulary candidates, the problem will be harder and it is impressive that URPE showed good performance even though the number increases.
> >
> > For sequence length evaluation, I think it can show the main problems on RPE and URPE solved that clearly, so adding the results can make this paper more concrete.

---

> > > ### Author Response · Authors · 2022-08-07
> > > **Thanks very much for the quick response!**
> > >
> > > We are encouraged and delighted to know that our response has addressed your concerns. As you advised, we have updated our paper and provided an ablation study on the length of input sequences to make our paper more concrete. Please refer to Section C.1 in the Appendix. We sincerely thank you again for your valuable feedback!

---

### Official Review · Reviewer_zNBv · 2022-07-10

**Rating:** 8
**Confidence:** 4
**Soundness:** 3 good
**Presentation:** 3 good
**Contribution:** 3 good

**Summary:**

In the paper the authors first presented theoretical proof that the Relative Positional Encoding (RPE) based Transformers are not universal function approximators, unlike the originally designed Absolute Positional Encoding (APE) based Transformers are. One primary reason for this is that most RPEs are placed inside the softmax in the attention module. The softmax operator always generates a right stochastic matrix. This restricts the network from capturing positional information in the RPEs and limits its capacity. The paper also conducted synthetic experiments to support this claim empirically. To overcome this limitation, the authors provided two sufficient conditions for RPE-based Transformers to achieve universal function approximation. With this results, the authors proposed a new attention module called Universal RPE-based (URPE) Attention. The transformers with URPE-based Attention, called URPE-based Transformers, are universal function approximators. Finally they presented experimental results to demonstrate the effectiveness of Transformers with the proposed URPE-based Attention.

**Questions:**

Longformer, Big Bird such models try to solve the sequence length limitation of APE-based transformers from a different angle. Is it possible to provide a comparison of the URPE-based Transformers with these architectures in terms of model size and performance?

**Limitations:**

I found the paper well rounded and a good addition to the Transformer literature. A more though comparison can help boost confidence on this work.

**Strengths And Weaknesses:**

Strengths:
As the authors noted that RPE-based Transformers generalize better on longer sequences compared to their APE-based counterparts. Transformers with RPE can achieve strong performance in language understanding and language generation tasks. RPEs are also popularly used in other domains to encode translation/rotation-invariant structural signals. Also, RPE makes Transformer easily be extended to other data modalities, such as image and graph, as the relative distance naturally preserves invariant properties for several important transformations like rotation and translation. Hence, there are many advantages of using RPEs and therefore these encodings became increasingly popular. But the authors pointed out one major limitation of RFE-based transformers. Transformers with RPE are not universal function approximators. That is, there exist continuous sequence-to-sequence functions that RPE-based Transformers cannot approximate no matter how deep and wide the neural network is. This paper not only pointed out this limitation, but also proposed Universal RPE-based (URPE) Attention to overcome this drawback. The authors presented theoretical proofs and experimental results to support their claims.

---

> ### Author Response · Authors · 2022-08-01
> **Response to Reviewer zNBv**
>
> Thank you very much for the careful review!  We would like to point out that Longformer/Big Bird that you mentioned and our work indeed study and improve different aspects of the Transformer model.
>
> Longformer, Big Bird, and many other seminal works, including Sparse Transformer[1], Linformer[2], Reformer[3], Performer[4], Random Feature Attention[5], and Transformer-XL[6], are in the family called "Efficient Transformer"[7]. As the name suggests, all the above models target improving the inference efficiency and reducing the computational/memory cost in the self-attention module, particularly for long sequence understanding and generation tasks. However, our work investigates the model capacity by studying whether the Transformer model can approximate continuous functions well.
>
> As those models and ours target solving different issues (i.e., efficiency v.s. capacity) in the Transformer architecture, they can be well combined. This is precisely what we try to deliver in the URPE-based Transformer-XL experiment: The Transformer-XL model can be improved with URPE-based attention for long sequence generation tasks. We hope our explanation can address your concerns and will conduct more experiments for URPE-based attention with other efficient models.
>
> [1] Child, Rewon, et al. "Generating long sequences with sparse transformers." ICML 2018.
>
> [2] Wang, Sinong, et al. "Linformer: Self-attention with linear complexity." arXiv preprint 2020.
>
> [3] Kitaev, Nikita, Łukasz Kaiser, and Anselm Levskaya. "Reformer: The efficient transformer." ICLR 2020.
>
> [4] Choromanski, Krzysztof, et al. "Rethinking attention with performers." ICLR 2021.
>
> [5] Peng, Hao, et al. "Random feature attention." ICLR 2021.
>
> [6] Dai, Zihang, et al. "Transformer-xl: Attentive language models beyond a fixed-length context."  ACL 2019.
>
> [7] Tay, Yi, et al. "Efficient transformers: A survey." ACM Computing Surveys (CSUR) (2020).

---

### Official Review · Reviewer_gvHj · 2022-07-10

**Rating:** 8
**Confidence:** 4
**Soundness:** 4 excellent
**Presentation:** 4 excellent
**Contribution:** 4 excellent

**Summary:**

The paper presents an analysis which shows that transformers with relative positional encoding are not universal sequence to sequence function approximators. This is shown rigorously but also intuitively based on the fact that traditional attention will sum to 1 hence given the same inputs the output is always going to be the same. The paper further shows that if an attention function satisfies two conditions then the resulting transformer model is a universal approximator for sequence to sequence functions. Based on this analysis the authors propose a simple and easy to understand modification to the attention function and show experimentally that 1) the resulting transformer can identify absolute positions in the sequence and 2) it improves upon the traditional attention in a variety of real world tasks.

**Questions:**

My questions as mentioned in the weaknesses section above are mostly regarding the comparison to combining an absolute positional encoding and traditional RPE or even fixed RPE like ALiBi. Why would this approach be preferred? Have you experimented with adding APE together with RPE in your baselines?

Finally, I would also like to know about possible memory and computational cost increase with URPE.

**Limitations:**

There were no obvious limitations that needed to be addressed.

**Strengths And Weaknesses:**

Strengths
------------

- The theoretical assessment of the limitations of RPE is novel, interesting and very simple to follow
- The proposed universal RPE is also interesting and simple to follow
- I particularly enjoyed both synthetic tasks that clearly showcase that RPE cannot calculate the absolute positions of tokens while universal RPE can
- The experimental evaluation showcases that this simple proposed improvement can improve the results in various tasks

Weaknesses
---------------

- There is very little comparison to using both RPE and absolute positional encoding. For instance the synthetic tasks could all be solved using absolute positional encoding and although I understand that the point of these experiments is to showcase the universality of URPE the question remains for other tasks like language modeling for instance.
- Similarly there is little intuition and understanding that can be drawn from the paper regarding the need for absolute positions. For instance, why would absolute positions be needed for language modeling? All things being equal, does it change anything if a certain subdocument is between positions 100-150 vs 50-100? How can we know if the improvement actually comes from the increased expressivity or improved learning dynamics?
- Although the number of parameters is increased only marginally, there is little mention wrt the possible increase in computational and memory cost. Element-wise multiplication is cheap but it does require another activation map the same size as the attention to be kept in the accelerator's memory.

---

> ### Author Response · Authors · 2022-08-01
> **Response to Reviewer gvHj (2/2)**
>
>
> **Regarding the computational cost.** We further conduct memory and time costs profiling experiments on our URPE-based Transformers. We choose the vanilla Transformer as the backbone model. The number of layers and the hidden dimension are set to 12 and 768 respectively. The number of attention heads is set to 12. The batch size is set to 32. We vary the sequence length from [128, 256, 512]. We run profiling of all the models on a 16GB NVIDIA Tesla V100. Following Combiner [2], we compare the inference speed and memory costs of the vanilla Transformer with RPE and our URPE. The results are presented in Table 1 and 2, which show that our URPE only increases minor computational costs.
>
> |                                                              |         |         |         |
> | :----------------------------------------------------------- | :------ | :------ | :------ |
> | **Inference Runtime (ms in log base 2)**                                        | **128** | **256** | **512** |
> | RPE-based Transformer                                        | 4.55      | 5.60      | 6.79      |
> | URPE-based Transformer                                       | 4.59      | 5.66      | 6.91     |
> | Table 1. Inference Runtime (ms in log base 2) of RPE-based Transformer and URPE-based Transformer with different sequence lengths. |         |         |         |
>
> |                                                              |         |         |         |
> | :----------------------------------------------------------- | :------ | :------ | :------ |
> | **Memory (GB)**                                              | **128** | **256** | **512** |
> | RPE-based Transformer                                        | 0.96      | 1.12      | 1.86      |
> | URPE-based Transformer                                       | 0.97      | 1.17      | 2.04      |
> | Table 2. Peak memory usage (GB) of RPE-based Transformer and URPE-based Transformer with different sequence lengths. |         |         |         |
>
> [1] Bao, Hangbo, et al. "Unilmv2: Pseudo-masked language models for unified language model pre-training." International Conference on Machine Learning. PMLR, 2020.
>
> [2] Ren, Hongyu, et al. "Combiner: Full attention transformer with sparse computation cost." Advances in Neural Information Processing Systems 34 (2021): 22470-22482. https://openreview.net/forum?id=MQQeeDiO5vv&noteId=-h5HnwArwV-

---

> ### Author Response · Authors · 2022-08-01
> **Response to Reviewer gvHj (1/2)**
>
> Thank you very much for supporting our work! We respond to your questions as below.
>
>
> **Regarding comparisons between RPEs and absolute positional encoding (APEs) in the synthetic experiment.**
> It is correct that the APE-based Transformer can perfectly solve the designed synthetic tasks, and we have empirically verified this before the submission. As this experiment is a bit beyond the scope of our work (studying the capacity of RPE-based models since in many practical scenarios, e.g., long sequence, image, graph, APE is not straightforward to apply),  we purposely removed it from the paper to avoid confusion. We are willing to add it back if the reviewer feels it can strengthen our work.
>
> **Regarding combinations (and comparisons) of RPEs and APEs in real experiments.**
> Thanks for the question. Following your suggestion, we conduct experiments on language pre-training to test different PE strategies. We chose this task since we noticed that in some competitive pre-training methods like UniLMv2[1], APE and RPE have already been used together. We mainly test three model variants: APE+RPE Transformer, RPE Transformer, and our URPE Transformer. For all the models, RPE is set to the T5 version, following UniLMv2. We roughly keep the number of parameters of different models to the same and train the models in the BERT-base setting using the same hyper-parameters.
>
> Due to the tight schedule of the rebuttal period, we only obtained the validation loss in the pre-training stage (masked language modeling loss after 1M iterations on a hold-out validation set). We observed that the validation losses of the APE+RPE/RPE/URPE Transformers are 1.86/1.94/1.87, respectively. The results show that URPE Transformer is competitive with APE+RPE Transformers and is much better than RPE Transformers.
>
> Together with the above observations on the synthetic dataset, we can see that URPE is competitive/superior to previous APE/RPE or their combinations. We can add those results to the paper in the next version of the paper.
>
> **Regarding whether the improvement comes from the increased expressiveness or not.**
> Thanks for the question. Showing where the improvement comes from in a rigorous way is challenging. For the synthetic dataset, the task is designed to be difficult for RPE Transformer by theory. Therefore we believe the improvement is coming from better expressiveness. For the language pre-training task, the community usually observes that models with larger capacity get better results (e.g., GPT-2 v.s. GPT-3, BERT-base v.s. BERT-large). Therefore, in the experiment above, we think the improvement from RPE to URPE may also come from the power of better expressiveness. We agree it is an important question and will investigate it deeper.
>
> **Regarding why the community uses APE.**
> Thanks for the question. We think the language pre-training experiment above can answer it. It can be seen from the results that using RPE only is worse than using APE+RPE, which suggests that RPE may not be powerful enough to replace the APE module entirely. We agree with the reviewer that APE may not be a perfect way to model sequential behavior. Our work can be considered as an initial exploration to investigate the disadvantage of RPE and address its limitation.

---

### Official Review · Reviewer_TbAW · 2022-07-11

**Rating:** 7
**Confidence:** 4
**Soundness:** 3 good
**Presentation:** 3 good
**Contribution:** 3 good

**Summary:**

This paper presents URPE, a new universal relative position embed for stronger transformer architecture.
Starting from "absolute position embed-based transformers are universal approximators of continuous
sequence-to-sequence functions on a compact domain, authors analyze the approximation power of relative
position embedding in mathematics, and conclude that RPE-based transformers are not universal
approximators. Then authors propose a new relative position embedding with simple but effective
improvement to achieve universal representations. Experiments on both language and graph datasets show
URPE-based transformer’s advantages. Detailed ablation experiments and analyses make the improvements of
URPE reliable.

**Questions:**

See Weaknesses.

**Limitations:**

The authors briefly discussed their limitations in L348-350.

**Strengths And Weaknesses:**

### Strengths
1. The motivation and method of this paper are both proven in mathematics.
2. The proposed URPE is easy to implement and can adapt to any RPE-based transformers for further
improvements. It only introduces a few extra parameters to the original RPE.
3. This paper is well written and organized. Motivations, methods, and concerns in experiments are all very
clear and easy to follow.

### Weaknesses
1. Improvements seem to be marginal.
2. Experiments in each table are conducted under a single transformer architecture(e.g., Transformer-Base in
Tab.). It’s interesting to see how many gains URPE can obtain with all kinds of transformer architectures
(e.g., Transformer-Tiny/Small/Large, etc.).
3. Experiments in Vision Transformer. Relative Position Embedding is also widely used in many vision
transformers (e.g., Swin-Transformer). The results of the URPE-based vision transformer are important to prove
URPE’s generalization to other modalities.

---

> ### Author Response · Authors · 2022-08-01
> **Response to Reviewer TbAW**
>
> Thank you very much for supporting our work! We appreciate your advice on the experiments. Here are our responses to your questions:
>
> **Regarding the performance improvements.** It is worth noting that all the improvements are obtained with negligible more parameters compared to the backbone Transformers. For the Language Modeling task, there are only 4K newly introduced parameters while we obtain 0.8 lower test perplexity score. Besides, we can see that our URPE-based Attention enables the Graphormer to reduce more than 40\% relative MAE on the ZINC dataset, which is a significantly large performance gain. On PCQM4M dataset, our URPE-based Attention improves the performance of the Graphormer with 12.5M parameters to match the performance of the Graphormer with 48.3M parameters. Under the quantum chemistry precision, this improvement should be considered to be significant.
>
> **Regarding the model architectures and sizes.** As stated in Line 216 to 218, the principles of our experimental design include covering typical RPE-based Transformer architectures and sizes, which indeed aligns with your advice. Briefly, we choose three different architectures: 1) Transformer with T5-style RPE (in Section 5.1); 2) Transformer-XL (in Section 5.2); 3) Graphormer (in Section 5.3). Besides, the model sizes also vary from 12.5M to 151M, covering Transformer-Tiny, Transformer-Small, and Transformer-Base. We follow your advice and conduct experiments on the Language Modeling tasks with 4-layer and 8-layer Transformer-XL models. The results on the wikitext-103 dataset are presented in Table 1. Due to the time limitation and restrictions of computational resources, we will conduct experiments on models of large size and vision tasks and add the new results to the next version of our paper.
>
>
> |                                                              |         |         |
> | :----------------------------------------------------------- | :------ | :------ |
> | **Valid PPL**                                                | **L=4** | **L=8** |
> | RPE-based Transformer-XL                                     | 29.61   | 25.98   |
> | URPE-based Transformer-XL                                    | 28.72   | 25.15   |
> | Table 1. Validation Perplexity of RPE-based Transformer and URPE-based Transformer with different number of layers. |         |         |

---

> > ### Comment · Reviewer_TbAW · 2022-08-08
> > **Post-rebuttal**
> >
> > Thanks for the authors' responses. Overall, I think this is a practical method with good theoretical proof. The paper writing, mathematical analysis, and experiments on kinds of modality make this paper solid and reliable. I choose to maintain my original positive rating.

---

> > > ### Author Response · Authors · 2022-08-09
> > > **Thanks very much for the response!**
> > >
> > > We sincerely thank you for your appreciation of our work! Your feedback is insightful to help us improve our paper. Thanks!

---

### Author Response · Authors · 2022-08-07
**General Response: New Results & Paper Updates**

We sincerely thank all the reviewers and the area chair for their efforts in reviewing our paper. The comments have enlightened us to ponder how to improve the quality of our submission. As you advice, we add new experimental results and discussions to our paper, including:

- The Runtime and Memory Usage Evaluation of our URPE-based Transformer (Section C.4 in the Appendix)
- The performance of our URPE-based Transformers of different model sizes (Section C.5 in the Appendix)
- The comparison between our URPE-based Transformers and Transformers with both APE and RPE (Section C.5 in the Appendix)
- Ablation Study on the length of input sequences (Section C.1 in the Appendix).

Please let us know if you have any further concerns and we are willing to answer any further questions you have on our paper. Thank you again for your insightful feedback.

Thanks!

Paper 1737 Authors

---

### Meta-Review · Area_Chair_Z7po · 2022-08-26

**Recommendation:** Accept
**Confidence:** Certain

**Metareview:**

This paper studies relative positive embeddings based Transformers. The authors present a negative result that there exist continuous sequence-to-sequence functions that relative based Transformers cannot approximate (irrespective of the depth and width of the network). The authors then propose a novel attention module, called Universal RPE-based (URPE) Attention which resolves this problem and show superior performance on a wide range of applications. There is a strong consensus amongst the reviewers that the paper is technically-solid, novel, well-motivated and has good practical applications. I agree with the reviewers and recommend acceptance.

**Award:**

No

---

### Decision · Program_Chairs · 2022-09-14

Accept